# Education for Sustainable Development (ESD) in Romanian Higher Education Institutions (HEIs) within the SDGs Framework

**DOI:** 10.3390/ijerph19041998

**Published:** 2022-02-11

**Authors:** Alexandru Sebastian Lazarov, Augustin Semenescu

**Affiliations:** 1Engineering and Management, Industrial and Robotics Engineering, University Politehnica of Bucharest, 313 Splaiul Independentei, 060032 Bucharest, Romania; augustin.semenescu@upb.ro; 2Academy of Romanian Scientists, 3 Ilfov Str., 050044 Bucharest, Romania; 3American Romanian Academy of Arts and Sciences, P.O. Box 2761, Citrus Heights, CA 95611-2761, USA

**Keywords:** sustainability, barriers, institutional framework, campus operations, education, research, outreach and collaboration, on-campus experiences

## Abstract

The purpose of this research is to explore barriers and challenges that HEIs in Romania must overcome in order to incorporate education for sustainable development (ESD), but also to provide key recommendations regarding factors that can facilitate sustainability in higher education in Romania. In terms of the approach, a qualitative approach was used consisting of semi-structured interviews involving seven actors from Romanian HEIs, actors who are in charge of ESD adoption. All respondents were in charge of ESD adoption within the Romanian universities that were part of the analysis. The research outlined that the analyzed Romanian universities have implemented ESD, but in terms of planning and practices, they have taken isolated actions. Moreover, major barriers and challenges have been highlighted, such as: funding deficiencies, a lack of experienced officers to adopt SD, change difficulties and improper public policies. The originality of the article resides in the fact the approach is holistic, which adds value to the literature in the area, particularly since, so far, research on ESD in Romanian higher education has focused only on particular factors.

## 1. Introduction

Sustainable development first became a key element with the publication of the Brundtland Report [1] in 1987. Closely connected to this report, which triggered the international awareness on sustainable development, was also the Paris Climate Change Agreement from 2015. This agreement prompted scholars to acknowledge, at an international level, that sustainability was endangered and from there the urgency for sustainable measures emerged. From this point forward, education came into focus. Thus, in 2002, the United Nations General Assembly (UNGA), declared the “United Nations Decade of Education for Sustainable Development (UN-DESD, 2005–2014)” [2].

According to UNESCO (United Nations Educational, Scientific and Cultural Organization), “Education for Sustainable Development develops and strengthens the capacity of individuals, groups, communities, organizations and countries to make judgements and choices in favour of sustainable development. It can promote a shift in people’s mind-sets and in so doing enable them to make our world safer, healthier and more prosperous, thereby improving the quality of life. Education for sustainable development can provide critical reflection and greater awareness and empowerment so that new visions and concepts can be explored, and new methods and tools developed” [3] (p. 1).

The key aims of the UN ESD Strategy reveal the urge of a paradigm shift within HEIs from several points of view, some in the power of universities, other in the governmental power: learning, teaching, providing tools and materials, research and development, regional cooperation and regulations and policies [4,5,6]. 

More recently, ESD was integrated in the 2030 Agenda [7]. Goal 4.7 of the 2030 Agenda refers particularly to Quality Education under SDG4 and can be seen in Figure 1. Even if the focus of the current article is strictly on SDG4, it must be stated that ESD brings a major contribution to all 17 SDGs listed in Figure 1 below:

In consonance with this goal, governments should guarantee that “all learners acquire knowledge and skills needed to promote sustainable development, including among others through education for sustainable development […]” [8]. Additionally, the Council of the European Union advised towards mainstreaming “the ambitions of the UN Sustainable Development Goals, in particular within the SDG4.7, into education, training and learning, including by fostering the acquisition of knowledge about limiting the multifaceted nature of climate change and using natural resources in a sustainable way” [9] (p. 5).

Consequently, the academic community became more and more focused on the critical role played by HEIs in the adoption of education for sustainable development (ESD), as a means to facilitate the transition toward sustainable development [10,11,12,13]. 

In this vein, ESD means the complete transformation of higher education, with the focus on various issues, such as: learning content and environment, pedagogy and outcomes. In order to continue the policies concerning ESD, in 2020, UNESCO adopted “Education for Sustainable Development: Towards achieving the SDGs” or “ESD for 2030” [14], which continues and updates the previous ESD documents; although, so far, the pandemic has postponed the launch of the new framework. “ESD for 2030” focuses on five priority actions, which can be found in Figure 2 below: 

In focusing on these priorities for diverse stakeholder groups, higher education institutions (HEIs) position themselves strategically at the nexus of research, development and industry, learning and practice, as well as education and community, bringing a major contribution in the effective implementation of the SD changes, meanwhile, also following the ESD roadmap for 2030, which outlines the necessity for conducting communication, outreach and advocacy activities [14]. 

On an international scale, in spite of the world education rankings in which European universities are prominent with some having attained SDG goals [15,16,17], the European continent is still behind Japan, USA and South Korea [18].

In accordance with HEIs’ attempts to prove their “commitment and better embed sustainability”, Lozano et al. [19] produced a review gathering results from a worldwide survey and, by consequence, divided their results into eight categories, according to Figure 3 [19]: 

This research, along with the above mentioned categories, represents the basis for the current article. At the same time, similar researches in various countries have been explored for a better understanding of the subject. In this respect, Farinha et al. (2020) [20] focused on how ESD has been implemented into programs at HEIs in Portugal. Their study focused on two periods: the DESD 2005–2014 as compared to 2018. In terms of findings, major improvements were observed between the two periods concerning ESD adoption. At the same time, the research pointed out that Portuguese public universities draw “their own strategies and policies on ESD, leading them to introduce initiatives arising from their proactiveness rather than governmental edicts” [20]. Moreover, in terms of drawbacks, the scholars found that the most common barriers that Portuguese HEIs faced were “lack of funding, not properly trained people and inept government policies” [20].

The results of studies such as that of Aleixo, Leal and Azeiteiro (2018) [21] have shown that the role played by HEIs in the promotion of sustainability is major, but only by overcoming both challenges and barriers could sustainability truly be achieved. The analysis explored the opinions of twenty stakeholders from four Portuguese HEIs, and the outcomes revealed that there was a lack of awareness concerning the concept of sustainable HEIs. Additionally, the authors suggested that due to the decrease in both student numbers and financial resources, which are actually closely inter-related, there were difficulties in implementing SD in HEIs. The research pleads in favor of the necessity of change for HEIs regarding (1) the identification of new financial resources, (2) an increased flexibility of HEIs, (3) inclusive strategies and goals, (4) increased engagement with internationalization and continuous learning practices, and (5) better HR management.

In Romania, in terms of ESD, some universities have signed the University Charter for Sustainable Development, known as the Copernicus Charter [22]. In a previous research focusing on the Romanian professor training curriculum in relation to the ESD competencies, it was revealed that this curriculum does not nurture ESD skills [23]. Furthermore, another research pointed out that the economic public higher education system in our country has taken small steps in ESD, but what stands as hard to change is mentality. In addition, a difference between public and private Romanian universities was found as the former appear more involved than the latter [24]. 

More recently, in line with the previously presented research, Piroșcă et al. [25], in their study, reported that Romanian students of one particular university from Bucharest have perceptions of ESD below average, which implies that ESD practices are currently at an early stage in Romania [25]. Another research [26] focused on the students’ opinions about ESD, also revealing barriers to sustainability. In their perception, they represent the most important obstacle in the achievement of ESD [26].

Another research [27] “maps and tracks the actual performance in quality education across the European Union member states (EU27), focusing on SDG4—quality education from Agenda 30” [27] (p. 1). According to this study, Romania stands as a low performer.

Therefore, in general, the focus has been on particular issues concerning ESD, either on professors or students, rather than on a more complex perspective. Starting from these above mentioned researches, the purpose of this study is to expand the perspective on the development of ESD in Romanian HEIs via an interview applied to seven public universities from Romania; more precisely, to the leaders in charge of the adoption of sustainability in the studied universities. The following research questions have been addressed in this article:

RQ1. Have Romanian universities adopted ESD in their policies, campus operations, education (sustainability incorporated curriculum/pedagogy), research, community outreach and collaborations?

RQ2. What are the barriers against ESD adoption within Romanian HEIs?

This research offers a new perspective regarding the adoption of ESD in Romanian HEIs, contributing to the literature by reflecting the manner in which they promote sustainability and also by revealing what barriers they face in ESD adoption. The article highlights the importance of assessing the adoption of sustainability in the Romanian educational sector, in order to establish a country profile for ESD implementation. Additionally, this study conducted in Romania, can be helpful for other HEIs both from Romania and from other countries to comprehend the policies and measures taken, with the final goal to encourage best practice in the adoption of ESD. 

## 2. Materials and Methods

### 2.1. Nature of the Research

For answering the research questions of the paper, the research consists of semi-structured interviews involving seven actors from Romanian HEIs, actors who are in charge of ESD adoption. Therefore, the methodology used is qualitative, having as its main aims to assess (1) the adoption of ESD and also (2) the barriers encountered by universities in its implementation. The researchers opted for this type of approach as it offers deeper understanding regarding respondents’ opinions [28]. In addition, the authors decided in favor of a qualitative research because they considered that the analysis will be enriched with the examples provided by the participants, whereas the quantitative analysis did not provide this alternative. This is the main reason why an interview that consisted of close-ended questions was used, as participants were able to add their comments to the answers.

### 2.2. Interview Script

At the basis of the interview script was the study of Lazano et al. [19] and the seven dimensions that they proposed (the background category was excluded) together with the challenges that scholars [20,21] discovered as far as ESD adoption is concerned. Therefore, the interview was divided into two parts, the first made up of the above mentioned six dimensions, and the second made up of the barriers and challenges that ESD adoption involves. Before running the interviews, the scripts were pre-tested with the help of PhD students and professors from Romanian HEIs, and, as a consequence, their recommendations were implemented in the scripts.

The interview was not very long, consisting of 50 close-ended questions (see Appendix A). Each question was followed by an opportunity according to which participants could comment or offer examples on the question already answered. The close-ended questions were of one type: “Yes/No/I don’t know/I don’t want to answer”.

The aim of the open questions was to encourage the respondents to discuss and, therefore, offer broad and enriching answers.

### 2.3. Study Population

The study group is made up of seven universities from Romania, the main objective being that of offering a general framework describing ESD adoption in Romanian HEIs. 

Although not all universities from Romania are included in this research, the participants were those in charge of SD implementation in their universities; therefore, their opinions are valuable for the context. Before proceeding with the interviews, each participant was contacted and a day was scheduled for each interview. 

### 2.4. Data Gathering and Analysis

The participants agreed on audio recording the interviews, which were further transcribed. Interviews were conducted between September and November 2021. Each participant received a codified identification (HEI1; HEI2…), which ensured confidentiality. Most interviews occurred via Zoom and rarely (2) via Skype. Three interviews had to be rescheduled. 

After being transcribed, the interviews were analyzed in a coding system in accordance with the approach used, that being descriptive statistics. The analysis was conducted using IBM SPSS 24, enriched with the examples provided by the participants. 

## 3. Results

The analysis started with the recognition of the ESD adoption by the Romanian HEIs that participated in the research. Thus, out of seven participants, five admitted that their HEI has signed the Copernicus Charter, which testifies for their awareness as far as SD is concerned.

Next, the seven categories concerning SD adoption within Romanian HEIs were analyzed. The first category referred to the institutional framework and the implementation of SD within it. Therefore, elements such as: SD policies, implementation of SD in the HEIs’ mission and goals, the existence of SD office and staff dedicated for these policies and the existence of an SD budget were all carefully analysed.

The answers about SD implementation within the institutional framework of HEIs included elements found in Figure 4, such as: SD adoption in vision and mission, goals and objectives (of HEIs), SD policies, SD strategic plans, staff members dedicated to SD, the office supporting SD implementation within the institution and budget for SD initiatives.

According to the participants, SD implementation within the *institutional framework* of HEIs has already been incorporated. No matter if it implies the adoption of SD in the mission, goals and objectives of universities, or simply the implementation of SD policies, the existence of these elements within Romanian HEIs proves that the commitment to SD is important in these institutions. In addition, respondents agreed that although there are SD offices and staff committed to SD adoption in most of the participant institutions, they are hardly enough and, therefore, more efforts should be made in order to develop these structures. Additionally, although an SD budget exists in five out of the seven HEIs, the interviewees agree that budgets are small. Moreover, most respondents consider that since SD budgets are key factors as far as sustainability is involved, measures should be taken in the future in order to increase these budgets.

In terms of the *campus operations*, the following actions were seen as the most important for the adoption of sustainability in each studied HEI: Digitalisation (100%);Waste reduction (100%);Access and facilities for disabled people (100%);Waste bins for separation (100%);Alternative energy (71%);Energy reduction (71%);Building operations (71%);Water management (43%).

Figure 5 provides a general framework of the campus operations in which SD was implemented:

Although some of the above mentioned measures include one another, the first key aspects that participants recognized were: digitalization, waste reduction (in fact by digitalisation universities reduce paper-based forms, which means also waste reduction), access and facilities for disabled people and waste bins for separation.

Some of the interviewees admitted that, during the last years, the adoption of SD measures in campus operations became more and more focused on social equality and plans to improve both energy efficiency and waste reduction.

As far as education is concerned, as Figure 6 proves, the most frequent measures implemented for SD were: Sustainability incorporated curriculum for all students (100%);Sustainability (optional) incorporated curriculum for all students (100%);SD major for Bachelor’s level (100%);SD major for Master’s level (100%);Role plays, simulation, discussions and debates on SD (100%);SD major for PhD’s level (86%);SD training for lectures/professors on SD (86%).

The participants confirmed that it is mainly professors and staff in leading teaching roles who usually go to training on the integration of SD into the educational curriculum. They offered as examples some educational programs focusing on SD and climate transformation.

Another category analyzed was research, which can be found in Figure 7. The adoption of the SD dimension in research consisted mainly of publications on SD (100%), the fact that professors and students are encouraged to conduct joint researches (100%), and also of green projects on SD that are implemented in the universities (86%).

The less frequent measures concerning research are the use of research generated in SD teaching and funding of SD research. 

Another category analyzed was that of the adoption of SD in outreach and collaboration. In this vein, a particular emphasis was put on: Collaboration in SD research projects (100%);SD partnerships with other stakeholders (100%);SD Exchange programs (86%);Joint SD degrees and research with other HEIs (86%).

These measures stand as the most frequent according to the participants’ opinions. Therefore, universities in Romania pay particular interest to research collaborations with different stakeholders and the community, often becoming involved in such partnerships. Not one single Romanian university is part of the UN Regional Centre of Expertise.

Next, the on-campus experience was analyzed. The answers revealed that two actions were the most acknowledged by the interviewees:Involvement of students in SD activities (100%);Involvement of professors and researchers in SD activities (100%).

Finally, the last part of the analysis focused on the barriers concerning the implementation of SD in Romanian HEIs. These barriers can be seen in Figure 8 below:

The most frequent barriers identified by the participants were: Deficiencies in (lack of) government policies (100%);Funding (100%);Lack of staff and experienced officers to adopt SD (100%);Resistance to change (100%).

Therefore, the lack of funds and the lack of (or deficiencies in) government policies were among the most important challenges faced by Romanian HEIs in SD adoption. Moreover, the lack of experienced staff, together with a high resistance to change were also main obstacles expected to hinder SD implementation in Romanian HEIs.

## 4. Discussion

The findings reveal that out of seven participants, five admitted that their HEI has signed the Copernicus Charter, which is important but not decisive for SD adoption. According to the participants, SD implementation within the institutional framework of HEIs was already incorporated. Thus, as HEI3 stated, his university promotes “the principles of ESD and environmental protection”, this being stipulated also in their strategy. Therefore, SD policies were implemented in the goals, missions and objectives of Romanian HEIs, proving that there is a high commitment to SD and awareness in these institutions. In addition, participants pointed out that, although an SD budget exists in five out of the seven HEIs, the budgets are small. As HEI5 pointed out, “The University has a budget for initiatives promoting SD, but it is very small, although there are constant struggles to increase this budget.” Additionally, HEI4 offered as examples the “lack of budget for training, for curriculum reform but also for on-campus initiatives”, related to SD practices.

In terms of the campus operations, several actions were seen as the most frequent for the adoption of SD: digitalisation (100%); waste reduction (100%); access and facilities for disabled people (100%); waste bins for separation (100%). Several interviewees admitted that, during the last years, the adoption of SD measures in campus operations became more and more focused on social equality and plans to improve both energy efficiency and waste reduction. Education was also analysed as far as SD adoption was concerned. A key factor was the sustainability incorporated in the curriculum but also SD majors for all types of degrees (Bachelor’s, Master’s and PhDs). In addition, role plays, simulation, discussions and debates on SD as well as training for professors were considered of major importance. Indeed, training would enhance professors’ skills and competencies concerning SD, as scholars [29,30] often point out. However, the participants admitted that it is mainly professors and staff in leading teaching roles who usually go to training on the integration of SD into the educational curriculum. In this vein, HEI1 stated that: “Usually those who undergo training in aspects concerning the integration of SD in educational programs are the vice rectors, vice deans and various directors of faculty departments”. In addition, participants offered as examples some educational programs focusing on SD and climate transformation. Among the international programs mentioned often (6) was “Horizon 2020”.

Another category analyzed was research, which means knowledge creation and innovation. The adoption of the SD dimension in research consisted mainly of publications on SD, the fact that professors and students are encouraged to conduct joint researches, but also of green projects on SD that are implemented in the universities. An example in this case was provided by HEI2, who commented on the fact that: “We collaborate with research institutions at an international level. They are not only from Europe but also from Asia, such as South Korea, Japan, Taiwan, China and others”.

As far as the adoption of SD in outreach and collaboration is concerned, the most frequent measures were: collaboration in SD research projects; SD partnerships with other stakeholders; SD exchange programs and joint SD degrees and research with other HEIs. Therefore, universities in Romania pay particular interest to research collaborations with different stakeholders and the community, often becoming involved in such partnerships, a positive aspect also supported by researchers [31,32]. HEI7 mentioned several collaboration partnerships with the entrepreneurial environment, dealing with knowledge creation and technology transfer: “There are many university–industry and university–industry–community collaborations occurring even now. Being a technical university, we collaborate with Microsoft, Transelectrica, Nuclearelectrica, Siemens, Renault and other companies”.

The positions adopted by Romanian HEIs are similar to those stipulated in international declarations concerning SD and also in the academic literature [20] on this area, which proves once more that there is a shift toward collaboration in Romanian universities. However, not one single Romanian university is part of the UN Regional Centre of Expertise.

The on-campus experience analysis revealed that two actions were the most acknowledged by the interviewees: the involvement of students and involvement of professors and researchers in SD activities. As far as green projects are concerned, several were mentioned, such as measures: (1) to minimize water consumption, (2) to use ecological light bulbs, (3) to collect waste separately for recycling, (4) to minimize the use of paper-based forms, and (5) to implement new digital systems for better connection of students, researchers and industry. Thus, HEI3 commented on this subject: “We develop many on-campus projects, which link sustainability research to the environment and to society, and, to mention just a few, these are: Cleantech, Interreg Danube Transnational Programme, Aspire”.

Finally, the most frequent barriers in SD adoption identified by the participants within Romanian HEIs were: deficiencies in (lack of) government policies; funding; lack of staff and experienced officers to adopt SD and resistance to change. Therefore, the lack of funds and the lack of (or deficiencies in) government policies were among the most important challenges faced by Romanian HEIs in SD adoption. Additionally, the lack of experienced staff together with a high resistance to change were also main obstacles expected to hinder SD implementation in Romanian HEIs. The resistance to change was confirmed also by previous researches [33,34] as an obstacle in SD adoption.

## 5. Study Limitations and Recommendations

The limitations of this research can be addressed by further researches. First of all, the focus of this research has been mainly on SD and Romanian HEIs, not on a particular higher university institution from Romania, or on a comparative perspective between Romanian HEIs and HEIs from other countries. Therefore, the interpretation of the findings can only be made from a Romanian perspective.

In addition, there were also some limitations concerning the methods used. Thus, the respondents were only seven, which means that, if the number was bigger, the gathered data would have been more reliable.

## 6. Conclusions

The current research aimed at widening the data on how SD has been adopted in Romanian universities, focusing on several key aspects such as: institutional framework; campus operations; education; research; outreach and collaboration; on-campus experiences; barriers in implementing SD in Romanian public universities. The study revealed that in spite of the pandemic, which negatively impacted all domains and developments, Romanian HEIs have made major steps in incorporating SD. 

However, some unbalances can be seen from one university to another, which proves that they have implemented SD according to their own strategies and policies without being nationally integrated. This brings forward the barrier of a lack of government policies and funding as far as SD is concerned. 

The research also focused on the challenges that Romanian HEIs face while implementing SD. These barriers should be addressed and solutions should be provided for each of them.

Future research directions could consist of formulating several recommendations for the Romanian HEIs in order to overcome the barriers for the successful adoption of SD in higher education. Moreover, case studies could be useful in order to analyze the steps taken by particular universities for the adoption of SD.

## Figures and Tables

**Figure 1 ijerph-19-01998-f001:**
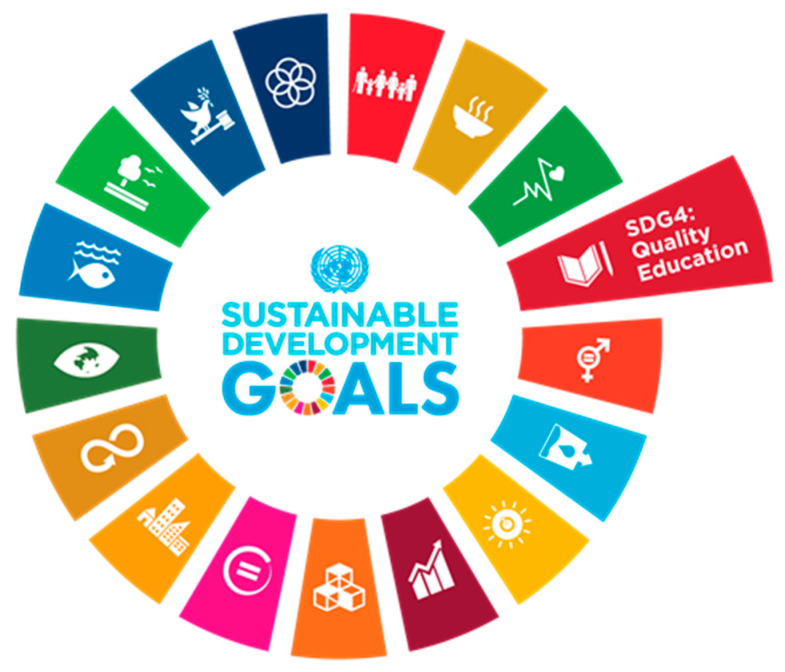
Agenda 2030. The Global Goals for SD. SDG4: Quality Education (Source: [3]).

**Figure 2 ijerph-19-01998-f002:**
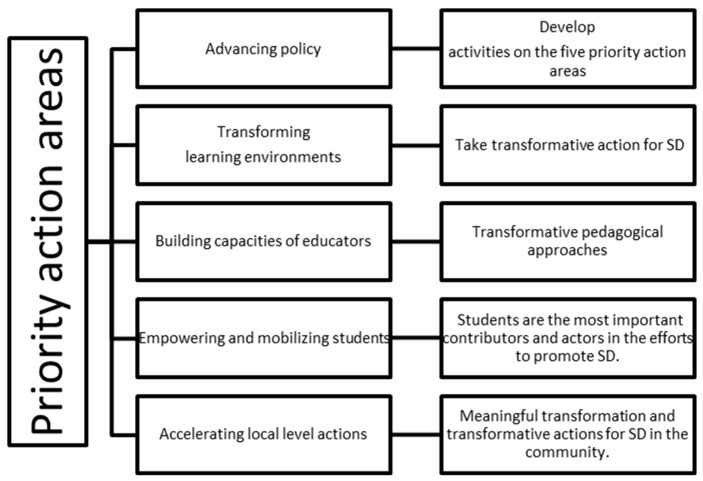
Priority action areas for “ESD for 2030” (Source: [14]).

**Figure 3 ijerph-19-01998-f003:**
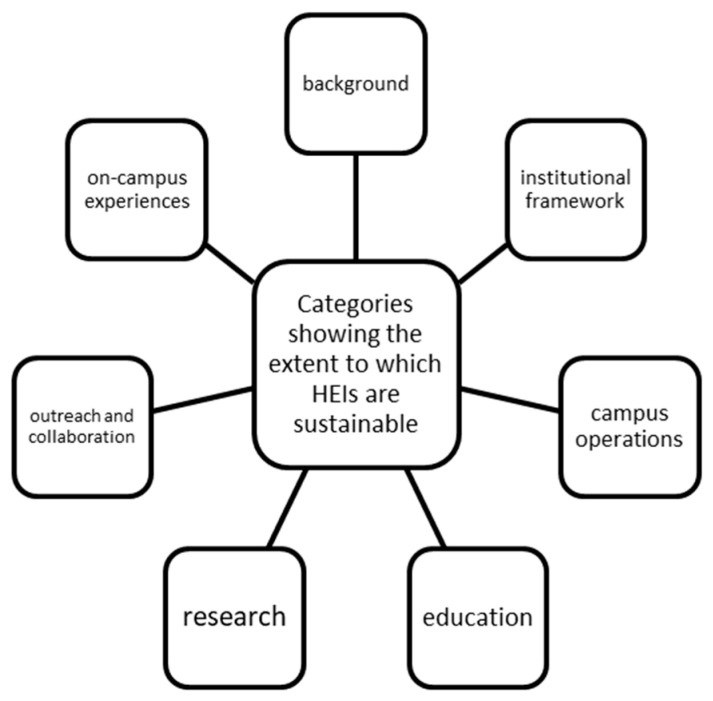
Categories showing the extent to which HEIs are sustainable (Source: [19]).

**Figure 4 ijerph-19-01998-f004:**
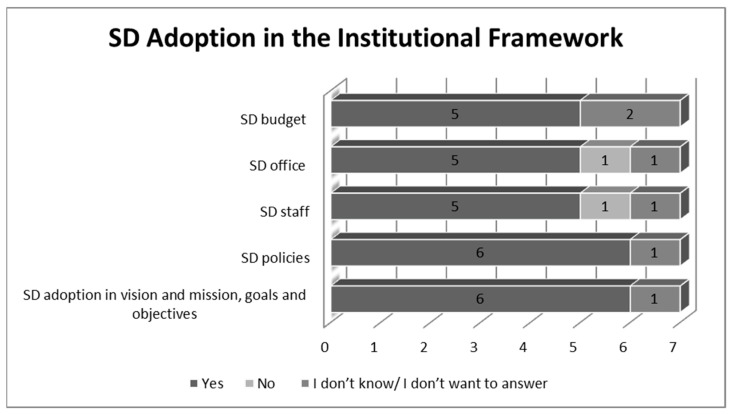
SD adoption within the institutional framework of HEIs (Source: the Authors).

**Figure 5 ijerph-19-01998-f005:**
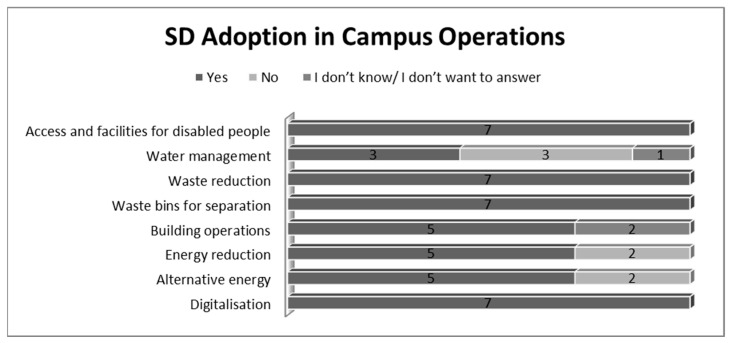
SD adoption in campus operations (Source: the Authors).

**Figure 6 ijerph-19-01998-f006:**
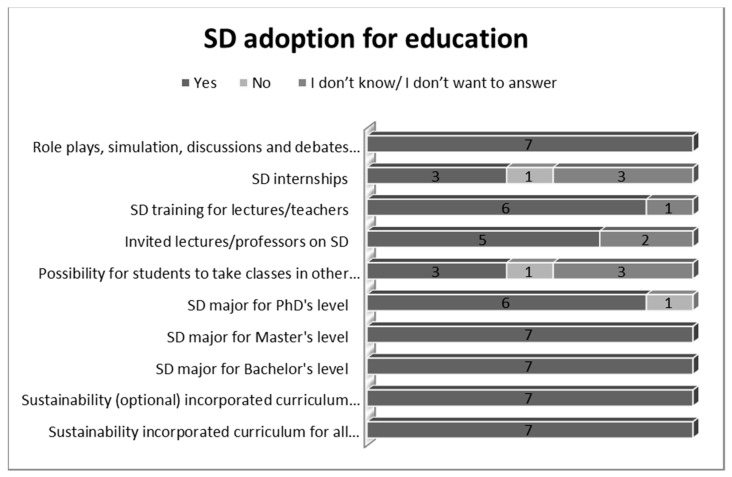
SD adoption in education (Source: the Authors).

**Figure 7 ijerph-19-01998-f007:**
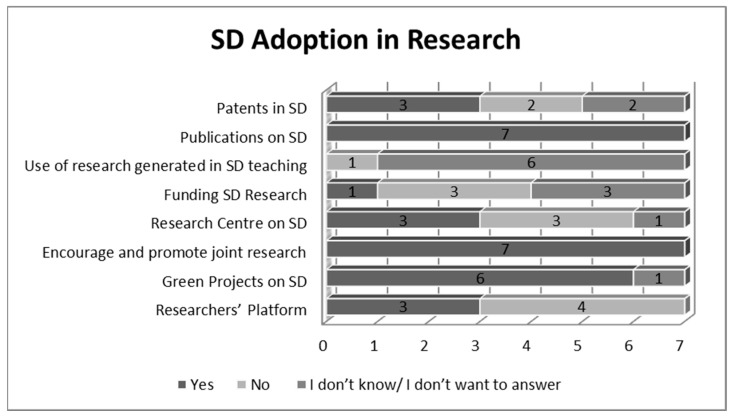
SD adoption in research (Source: the Authors).

**Figure 8 ijerph-19-01998-f008:**
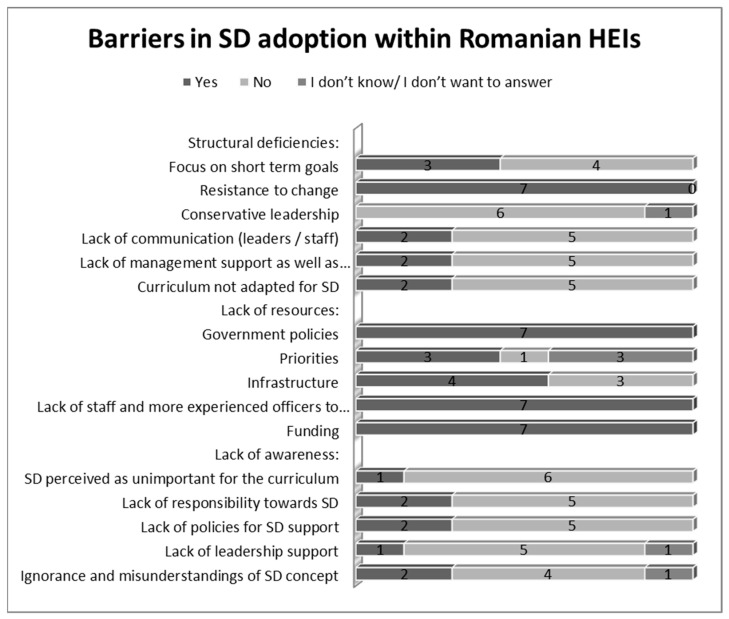
Barriers for SD adoption in Romanian HEIs (Source: the Authors).

## Data Availability

Not applicable.

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
