# Peer review of "Education for Sustainable Development (ESD) in Romanian Higher Education Institutions (HEIs) within the SDGs Framework"

_ijerph, 2022, doi:10.3390/ijerph19041998_

Round 1

Reviewer 1 Report

I am writing about the manuscript entitled Education for Sustainable Development (ESD) in Romanian Higher Education Institutions (HEIs) within the SDGs Framework. This paper has some noteworthy outputs and can contribute to the relevant literature. The authors however need to consider the below comments/suggested revisions to improve the paper.

  1. Keywords: The common practice in writing the keywords is to avoid the words used in abstract but the words listed as keywords are mostly used in abstract. Keywords should be 5-7 and avoid abbreviations.
  2. Introduction: The contributions of the study should be precisely highlighted.
  3. What are the sources of Figures? You might depict it below the figures.
  4. Some references are not properly cited in the text (see line 84, 98, 101, 106, 129)
  5. “addressed” instead of ad-dressed line 112
  6. “Future research directions…” should be at the end of Conclusions section instead of “Discussion” section.
  7. Please proof read the manuscript before submitting the revision.
  8. The final script of the interview should be included at the end as an appendix after Reference.

Author Response

1.Keywords: The common practice in writing the keywords is to avoid the words used in abstract but the words listed as keywords are mostly used in abstract. Keywords should be 5-7 and avoid abbreviations.

Thank you very much, your advice was implemented.

  1. Introduction: The contributions of the study should be precisely highlighted.

The contributions of the study were completed.

  1. What are the sources of Figures? You might depict it below the figures.

One figure was removed and for the rest sources have been added.

  1. Some references are not properly cited in the text (see line 84, 98, 101, 106, 129)

They were all properly corrected.

  1. “addressed” instead of ad-dressed line 112

This was corrected.

  1. “Future research directions…” should be at the end of Conclusions section instead of “Discussion” section.

They were moved.

  1. Please proof read the manuscript before submitting the revision.

Hopefully, no more mistakes are present.

  1. The final script of the interview should be included at the end as an appendix after Reference.

The final script of the interview was included at the end as an Appendix after Reference.

Thank you very much for your valuable time and input.

Reviewer 2 Report

Dear Authors,

Although the subject developed here, is discussed and relevant nowadays, it does not present significance originality.

Perhaps because a copy of the work of Lozano and other researchers is presented, and because the results based on only 7 interviews are presented as truths that are not properly documented and presented (please see this one): “This proves that there is a good communication between top management and the rest...”

In addition, it needs a major review (English review, review of the bibliographic references that follow this investigation and detailed review of the discussion and results), and the insertion of notorious bibliographical references on this topic that has long been developed in many countries, namely in Portugal.

After the suggested revisions, perhaps the article can be accepted. Under the proposed terms, I do not agree that it can be accept before this changes and corrections.

  • Review in the article the extensive transcripts that should be replace by the authors' sentences.
  • In the Portuguese context, there are several works that should be read in order to improve this article, for example Aleixo et al. Furthermore, it is suggest to read other works by Lozano in order to increase the results of the article.
  • The figure 1 must subject to revision. Was the figure built by the authors or was it copied?
  • The figures do not always help, in appealing terms, in the presentation of the article, please review and choose the figures that are really necessary.
  • It should indicate, in section 2, on which references the investigation is based.
  • On page 5 it is referred to Ramos et al and this reference should be corrected. This is not the reference that the study is based on, correct? There is no bibliographical reference to Ramos et al.
  • There is no reference to the questions that were presented in the 7 HEIs. Please indicate what the questions were.
  • The results and discussion should be review, namely as stated in the following: “This proves that there is a good communication between top management and the rest...”
  • Who, in each HEI, was the person responsible for answering the interview? A teacher, an officer or another stakeholder?

Author Response

  1. In addition, it needs a major review (English review, review of the bibliographic references that follow this investigation and detailed review of the discussion and results), and the insertion of notorious bibliographical references on this topic that has long been developed in many countries, namely in Portugal.

Both the English and the bibliographic references were reviewed. The discussion and results were also carefully looked into. They were also completed. In addition, new bibliography was added, particularly studies on the topic from Portugal.

  1. Review in the article the extensive transcripts that should be replace by the authors' sentences.

There have been added transcripts in the body of the article.

  1. In the Portuguese context, there are several works that should be read in order to improve this article, for example Aleixo et al. Furthermore, it is suggest to read other works by Lozano in order to increase the results of the article.

His works have been added and also other from the Portuguese literature. 

  1. The figure 1 must subject to revision. Was the figure built by the authors or was it copied? The figures do not always help, in appealing terms, in the presentation of the article, please review and choose the figures that are really necessary.

Figure 1 was deleted and sources have been added to the remaining ones.

  1. It should indicate, in section 2, on which references the investigation is based. On page 5 it is referred to Ramos et al and this reference should be corrected. This is not the reference that the study is based on, correct? There is no bibliographical reference to Ramos et al.

I am very sorry for this inconvenience. I must have read some material by Ramos et al. but did not add it to the bibliography. However the article is based on Aleixo et al. ‘s work and also on other works which have been specified in section 2.

  1. There is no reference to the questions that were presented in the 7 HEIs. Please indicate what the questions were.

The questions were added in the Appendix, after References.

  1. The results and discussion should be review, namely as stated in the following: “This proves that there is a good communication between top management and the rest...”

This particular phrase has been changed. Also the results and discussion section was detailed and improved.

  1. Who, in each HEI, was the person responsible for answering the interview? A teacher, an officer or another stakeholder?

In the Appendix was added the position of each participant in their HEI.

Thank you very much for your valuable time and input.

Reviewer 3 Report

  1. The authors should argue why they used an interview that consisted of  50 close-ended questions. This seems more like a questionnaire and not interview.
  2. The discussion section includes a summary of the results and little explination and analysis. The authors should rewrite the discussion section to focus more on the analysis of the results. Use of literature should be richer. 
  3. A section named 'Study limitations and recommendations' should be added at the end of the paper. 

Author Response

1.The authors should argue why they used an interview that consisted of 50 close-ended questions. This seems more like a questionnaire and not interview.

In addition, the authors decided in favor of a qualitative research because they considered that the analysis will be enriched with the examples provided by the participants, whereas the quantitative analysis did not provide this alternative. This is the main reason why it was used an interview that consisted of close-ended questions, as participants were able to add their comments to the answers.

  1. The discussion section includes a summary of the results and little explination and analysis. The authors should rewrite the discussion section to focus more on the analysis of the results. Use of literature should be richer.

The discussion section was rewritten and examples from the interviews were added. Also, the literature review was expanded and new sources were therefore added.

  1. A section named 'Study limitations and recommendations' should be added at the end of the paper.

This new section was added:

  1. Study limitations and recommendations

The limitations of this research can be addressed by further researches. First of all, the focus of this research has been mainly on SD and Romanian HEIs not on a particular higher university institution from Romania, or on a comparative perspective between Romanian HEIs and HEIs from other countries. Therefore, the interpretation of the findings can only be made from a Romanian perspective.

In addition, there were also some limitations concerning the methods used. Thus, the respondents were only seven, which is means that, if the number was bigger, the gathered data would have been more reliable.

Thank you very much for your valuable time and input.

Round 2

Reviewer 2 Report

Dear Authors,

The article presents some improvements that go along with what was suggested.

I consider, however, that the article is quite similar to that of Lozano et al. (2015), however I consider that the questionnaire is not well constructed and has not been properly validated based on the literature.

Although it has not been subject to a thorough review, it can be accepted under the proposed terms.

Sincerely,

Reviewer 3 Report

Thank you for making the changes recommended.